# EMBEDDING IMPROVES NEURAL REGULARIZERS FOR INVERSE PROBLEMS

## ABSTRACT

Obtaining meaningful solutions for inverse problems has been a major challenge with many applications in science and engineering. Recent machine learning techniques based on proximal and diffusion-based methods have shown some promising results. However, as we show in this work, they can also face challenges when applied to some exemplary problems. We show that similar to previous works on over-complete dictionaries, it is possible to overcome these shortcomings by embedding the solution into higher dimensions. The novelty of the work proposed is that we **jointly** design and learn the embedding and the regularizer for the embedding vector. We demonstrate the merit of this approach on several exemplary and common inverse problems.

## 1   INTRODUCTION

The solution of inverse problems entails the estimation of a model (solution) based on measured data, which is often noisy and ill-posed in nature (Tarantola, 1987; Parker, 1994; Kaipio & Somersalo, 2004; Tenorio et al., 2011). These challenging problems arise in diverse fields such as geophysics Tarantola (1987), astronomy, medical imaging (Nagy & Hansen, 2006), and remote sensing (Vogel, 2002). Given the ill-posed nature of the considered problems and the presence of noisy data, the application of regularization becomes essential to achieve a stable and meaningful estimate of the model. Conventional regularization techniques involve using specific functions tailored to acquire desired properties, like Total-Variation (Rudin et al., 1992) or $\ell_2$ smoothness (Tenorio et al., 2011). Alternatively, some methods leverage a-priori estimates of the model statistics, such as Gaussianity (Tarantola, 1987; Kaipio & Somersalo, 2004).

The art and sophistication in solving an inverse problem is to balance between the *prior* knowledge about the solution and the *likelihood*, that is, the data fit of the predicted solution. The solution is derived as a combination of regularization and data-fitting functions, and it represents a compromise between the data fit and the prior. Traditionally, the prior is perceived to be less credible than the likelihood, as the likelihood is directly tied to the data of the problem to be solved.

In recent years, machine learning has facilitated the development of data-driven regularization techniques and prior estimation. To this end, supplementary data is available, aside from the measured data, containing many plausible solutions to the inverse problem. This additional data is then utilized to learn a regularization procedure, aiming to achieve superior results compared to traditional methods. There are two main approaches for using available data for learning how to solve inverse problems. The *first* is problem specific, that is an end-to-end approach, where the regularization process is learned in the context of the particular inverse problem at hand. Such an approach was presented first in Haber & Tenorio (2003) and then significantly improved in learning proximal maps by Parikh et al. (2014); Mardani et al. (2018); Jin et al. (2017); Adler & Öktem (2017); Mukherjee et al. (2021) and reference within. The *second* is learning a prior independently of the inverse problem, and then using the prior for the solution of the problem. This approach has been proposed in several recent works that utilize diffusion models to learn the prior (Chung et al., 2022a;c;b). Nonetheless, regardless of the approach used to learn the regularization function, in all the considered methods, the regularization is applied to the solution directly, that is, in its original coordinates. In other words, the regularization function uses the original properties and landscape of the solution space to measure its goodness. Therefore, the landscape of the regularization function may be highly non-convex and "unfriendly" to optimization procedures, especially those that use

first-order methods such as gradient descent, with or without stochastic sampling nuances, such as Langevin dynamics. This is a well-known problem for optimization methods that operate on low dimensions, in the original solution space (see Tarantola (1987); Nocedal & Wright (1999)).

To this end, a technique that has been widely successful in the past was to embed the solution using an over-complete dictionary (see (Chen et al., 2001; Candes et al., 2006; Bruckstein et al., 2009) and references within). In this approach, one uses an over-complete dictionary and embeds the solution in higher dimensions than the original solution space. It is important to note that in this technique, the regularization is applied to the **embedded solution vector** rather than the original solution vector. Canonical methods use $\ell_0$ and $\ell_1$ norms for the regularization of the embedding vector. These techniques have been highly successful at producing plausible and meaningful solutions, whether the dictionaries were learned or predefined, even though the regularization function was very simple, like the $\ell_1$ norm. It is therefore interesting to understand if similar concepts can be applied to the latest inverse problem neural solution techniques, and how would they impact the obtained results.

**The contributions of this paper** are as follows: (i) We show how embedding based techniques can be derived and learned in the context of contemporary data-driven regularization techniques. We show that by learning the embedding dictionaries **and** the regularization function that operates on the embedded solution vector, one can obtain regularization functions that are "friendlier" to gradient-based optimization methods that are then utilized to solve the inverse problem at hand. (ii) Furthermore, we introduce two unrolled versions of the algorithm that can be interpreted as dynamical system in high dimensions that can bypass the highly nonlinear landscape of the problem in its original coordinates. Similar to other unrolled versions of an optimization process Adler & Öktem (2017) the unrolling allows for greater expressiveness and outperforms shared weights algorithms. (iii) By examining several common inverse problems, we demonstrate that common architectures and approaches that use the original coordinates of the solution, can be significantly challenged while embedding based techniques converge to meaningful solutions.

**Connection to prior work:** Our method can be viewed as an extension of two popular and separate lines of techniques proposed for the solution of inverse problems. The first is using over-complete dictionaries, which was proposed in Candes et al. (2006); Chen et al. (2001) and followed by many successful algorithms and implementations (see Bruckstein et al. (2009) and references within). Second, our work extends the incorporation of learning regularization priors Haber & Tenorio (2003); Parikh et al. (2014); Mardani et al. (2018); Jin et al. (2017); Adler & Öktem (2017); Mukherjee et al. (2021) by embedding the solution. For learning the embedding, existing algorithms seek to find the optimal embedding over-complete dictionary (see Aharon et al. (2006a;b); Horesh & Haber (2011); Kasiviswanathan et al. (2012); Agarwal et al. (2014); Huang et al. (2013) and references within), while such embedding was not used in the context of learning regularization. Our work combines and extends both approaches by **jointly** designing and learning an embedding and a regularization function in the high-dimensional embedded space.

## 2 MATHEMATICAL BACKGROUND AND MOTIVATION

We first introduce the required mathematical background, followed by a simple, yet important example that demonstrates the shortcomings of existing inverse problem solution methods in deep learning frameworks.

**Problem formulation.** We consider the estimation of a discrete *model* $\mathbf{x} \in \mathbb{R}^N$ from the measured data $\mathbf{b} \in \mathbb{R}^M$, where typically $M < N$, and the relation between $\mathbf{x}$ and $\mathbf{b}$ is given by

$$\mathbf{A}(\mathbf{x}) + \boldsymbol{\epsilon} = \mathbf{b}. \tag{1}$$

Here the forward mapping $\mathbf{A} : \mathbb{R}^N \to \mathbb{R}^M$ can be either linear or nonlinear. For simplicity now consider linear inverse problems. The vector $\boldsymbol{\epsilon}$ is a noise vector that is assumed to be Gaussian with 0 mean and $\sigma^2 \mathbf{I}$ covariance. The forward mapping, $\mathbf{A}$, typically has a large effective null-space, which implies that there are infinitely many models $\mathbf{x}$ that correspond to the same data, $\mathbf{b}$.

**Traditional inverse problem solution methods.** We now provide a brief review of traditional estimation techniques for the model $\mathbf{x}$ given observed data $\mathbf{b}$, the forward mapping $\mathbf{A}$, and the statistics of the noise $\epsilon$. Let us first consider a Bayesian point of view for the recovery of the solution of the inverse problem. Assume that the model $\mathbf{x}$ is associated with a Gibbsian prior probability

density function $\pi(\mathbf{x})$ of the form

$$\pi(\mathbf{x}) \propto \exp\left(-R(\mathbf{x})\right). \tag{2}$$

Then, the posterior distribution of $\mathbf{x}$ given the data $\mathbf{b}$ can be written as

$$p(\mathbf{x}|\mathbf{b}) \propto \exp\left(-\frac{1}{2\sigma^2}\|\mathbf{A}\mathbf{x} - \mathbf{b}\|^2 - R(\mathbf{x})\right). \tag{3}$$

To obtain a solution (or a family of solutions), one may look at a particular procedure that uses the posterior. One popular approach is to use the Maximum A-posteriori (MAP) (DeGroot, 2005; Tarantola, 1987) estimate that maximizes the posterior by solving the optimization problem

$$\mathbf{x}_{\text{map}} = \arg\min \frac{1}{2\sigma^2}\|\mathbf{A}\mathbf{x} - \mathbf{b}\|^2 + R(\mathbf{x}). \tag{4}$$

The solution can be achieved by gradient descent iterations of the form

$$\mathbf{x}_{k+1} = \mathbf{x}_k - \alpha\left(\sigma^{-2}\mathbf{A}^\top(\mathbf{A}\mathbf{x}_k - \mathbf{b}) + \boldsymbol{\nabla}_{\mathbf{x}}R(\mathbf{x}_k)\right). \tag{5}$$

Alternatively, it is possible to sample the posterior with some statistical sampling technique. For instance, one can use Langevin dynamics (Pastor, 1994), to obtain a sampler of the form

$$\mathbf{x}_{k+1} = \mathbf{x}_k - \alpha\left(\sigma^{-2}\mathbf{A}^\top(\mathbf{A}\mathbf{x}_k - \mathbf{b}) + \boldsymbol{\nabla}_{\mathbf{x}}R(\mathbf{x}_k)\right) + \sqrt{\alpha}\mathbf{n}, \tag{6}$$

where $\mathbf{n} \in N(0, \mathbf{I})$ is a random variable. Also, we note that the use of Langevin dynamics is very popular in diffusion models (Yang et al., 2022; Croitoru et al., 2023).

The most common estimation or regularization approaches do not associate $R(\mathbf{x})$ with the log of the prior, and use Equation 4 with some desired properties of the solution such as rather low total-variation (Tenorio et al., 2011). By doing so, such traditional methods seek to balance between the prior and the likelihood. The regularization $R(\mathbf{x})$ is only approximately known, and in many cases is heuristic-based. Therefore, the solution $\mathbf{x}_{\text{map}}$ or the samples obtained via Langevin dynamics from Equation 6 represent a compromise between data fidelity (likelihood) that is obtained by minimizing $\|\mathbf{A}\mathbf{x}_k - \mathbf{b}\|_2^2$ and the prior incorporated by $R(\mathbf{x}_k)$. In particular, in most cases, for most traditional priors, the value of the prior probability at $\mathbf{x}_{\text{map}}$ is small. That is, the solution to the inverse problem would not be a likely solution if we consider the prior alone. Recent techniques seek regions of agreement between the prior and the likelihood. Recent advances in probability density estimation suggest that the regularization $R(\mathbf{x})$ can be estimated from data with greater accuracy compared to heuristic-based approaches such as TV priors, by utilizing a neural network (see (Yang et al., 2022; Croitoru et al., 2023) and references within). This is a paradigm shift. It implies that we seek solutions $\mathbf{x}$ that are significantly closer to the peak(s) of the prior, if they are to be realistic samples from the prior that also fit the data. As we see next, this makes the estimation of the model $\mathbf{x}$ substantially more difficult, because we need to derive algorithms that avoid local minima, and to find the global minima of the neural regularize $R(\mathbf{x})$.

We now provide an example that showcases our discussion above.

**Example 2.1. The duathlon problem.** *Consider a duathlon that is composed of cycling and running*

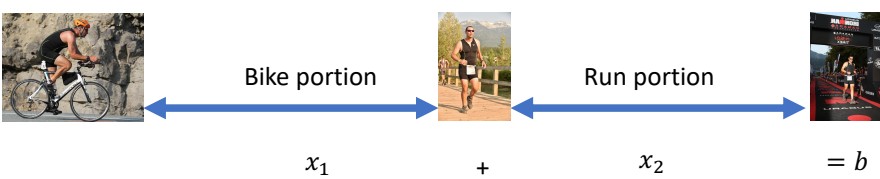

Figure 1: The duathlon problem, estimate the bike and run time from the total competition time.

*segments. Assume that we can measure the total time for an athlete to complete the duathlon, but we are unable to see the time it takes her to finish a particular segment. The question that we pose is, what was her time spent in each segment.*

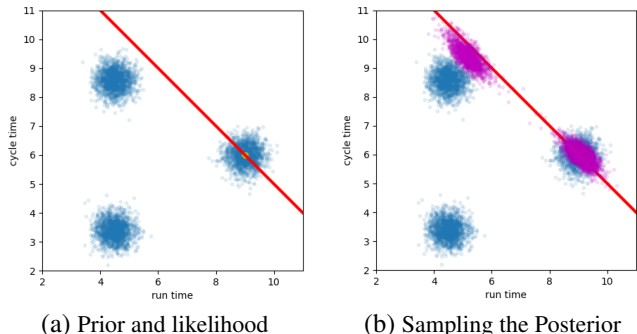

(a) Prior and likelihood      (b) Sampling the Posterior

Figure 2: Experiments with the dualthlon problem. The prior is made of three Gaussians and the data (yellow point) are presented on the left along with all possible points that fit the data (red line). Recovering a model given the data by sampling the posterior using diffusion is presented on the right (magenta points).

*The mathematical model for this problem is a simple single linear equation, as shown in Equation 1 with $\mathbf{A} = \begin{bmatrix} 1 & 1 \end{bmatrix}$ and $\mathbf{x} = [x_1, x_2]^\top$, where $x_1$ is the time spent on cycling and $x_2$ is the time spent on running. Our goal is: given the single equation $x_1 + x_2 + \epsilon = b$, where only $b$ is available, and $\epsilon$ is a Gaussian noise, estimate $x_1$ and $x_2$. Without any prior information, it is impossible to estimate $x_1, x_2$ given the data $b$. However, if we observe previous dualthlete data, we are able to estimate the distribution of the times $x_1$ and $x_2$. Such a distribution is plotted in Figure 2(a). In our example, the distribution is made of 3 groups. Assume that we measure a data point of an individual athlete (marked in yellow in Figure 2(a)). We can try and estimate the individual's cycling and running times $x_1, x_2$ by sampling from the posterior. To this end we use the algorithms discussed in Song et al. (2022); Chung et al. (2022b) that uses the prior within a diffusion process. The result of this approach is presented in Figure 2(b). The sampling from the posterior contains two main groups. However, while it is evident that one group is realistic (and indeed the true solution is sampled from this group) the second group is highly unlikely. This is because it does not coincide with the points of the prior. In fact, the points in this group are of very little probability to occur. The two groups we observe represent two local minima of the posterior where the one on the right is also the global minima. Nonetheless, starting at many random points, stochastic gradient-based optimization for the posterior is likely to converge to local both global and local minima, unless the Langevin dynamics is applied for a very long time. Furthermore, since diffusion-based algorithms typically avoid computing the probability and compute the score (that is, $\nabla \log \pi(x)$) instead, it is impossible to quantify the posterior probability of each point that is obtained by a diffusion process.*

*We thus observe that in this very simple case, diffusion models and other models based on optimization can yield unsatisfactory results. We have observed similar problems for much larger and more realistic problems, such as the estimation of susceptibility from magnetic data and image deblurring with very large point spread functions. For some of these problems, many local minima were observed and it was impossible to find samples that are close to the global minima.*

The problem above demonstrates two key shortcomings of existing approaches to solve inverse problems using deep learning frameworks, assuming sufficient reliable training data is available:

- The prior can be estimated well and we expect that the posterior will overlap some parts of the prior. Therefore, we seek points that are close to the **global** minima of the posterior.

- Finding a global minimum is a very difficult problem. Nonetheless, with the emergence of complex priors (such as diffusion models), such regularization leads to highly nonconvex problems where many local minima typically exist.

We now discuss a methodology that allows us to obfuscate these difficulties.

## 3 REFORMULATING THE SOLUTION OF INVERSE PROBLEMS BY EMBEDDING

As previously discussed, the main issue in solving the inverse problem, both in terms of sampling and in the MAP estimation is that we face a global optimization problem with many local minima. We now show that by reparametrizing the problem and embedding it in higher dimensions it is possible to obtain a more convex problem that is easy to work with and sample from, and therefore to find more likely solutions to the inverse problem.

### 3.1 HIGH DIMENSIONAL SOLUTION EMBEDDING

Let us consider an embedding of the solution $\mathbf{x} \in \mathbb{R}^N$ into a vector $\mathbf{z} \in \mathbb{R}^K$ where $N \ll K$, with an embedding matrix $\mathbf{E} : \mathbb{R}^K \to \mathbb{R}^N$, that is,

$$\mathbf{x} = \mathbf{Ez} \tag{7}$$

The vector $\mathbf{z}$ is the new variable we work with, and we solve the inverse problem with respect to $\mathbf{z}$ rather than $\mathbf{x}$. In what follows, we will learn an embedding and a regularization function that operates on $\mathbf{z}$ such that the resulting optimization problem is more attainable to numerical treatment compared with the original one for $\mathbf{x}$.

As we discussed in the introduction, the idea of embedding the solution in a larger space and regularizing the embedding vector has been thoroughly studied and discussed in the field of inverse problems (Candes et al., 2006) in the context of $\ell_1$ regularization and basis pursuit Chen et al. (2001). In this case, one replaces the original problem with the $\ell_1$ regularization

$$\mathbf{z}_{\mathrm{map}} = \arg\min \frac{1}{2\sigma^2}\|\mathbf{AEz} - \mathbf{b}\|^2 + \gamma\|\mathbf{z}\|_1 \tag{8}$$

which is associated with the prior density $\pi(\mathbf{z}) \propto \exp(-\gamma\|\mathbf{z}\|_1)$. The density in this case is log-concave and, hence, robust convex optimization algorithms can be used to solve the problem. The complicated part now is to choose an appropriate embedding matrix $\mathbf{E}$. As discussed in the introduction, the main focus of this line of work was to learn an appropriate embedding assuming the $\ell_1$ prior. An extension of Equation 8 is to **jointly** learn the embedding matrix $\mathbf{E}$ **and** a regularization function. Furthermore, we propose an unrolled version of this process that yields a neuro-ordinary differential equation (Haber & Ruthotto, 2017; Weinan, 2017; Chen et al., 2018).

### 3.2 LEARNABLE EMBEDDING AND REGULARIZATION IN HIGH DIMENSIONS

Equation 8 uses a high dimensional embedding, and employs the $\ell_1$ norm as a regularization for $\mathbf{z}$. However, one can learn a regularization function $\phi(\mathbf{z}, \boldsymbol{\theta})$, with parameters $\boldsymbol{\theta}$. This leads to a minimization problem of the form

$$\mathbf{z}_{\mathrm{map}} = \arg\min_{\mathbf{z}} \frac{1}{2\sigma^2}\|\mathbf{AEz} - \mathbf{b}\|^2 + \phi(\mathbf{z}, \boldsymbol{\theta}) \tag{9}$$

By carefully learning both $\mathbf{E}$ and $\phi$, we can obtain an optimization problem with favorable properties. This is motivated by the following theorem:

**Theorem 3.1.** *(Learning to optimize ). Let $\mathbf{x}_{\mathrm{map}}$ be the solution of the problem in Equation 4. Then, it is possible to choose an embedding $\mathbf{E}$ and a regularization function $\phi(\mathbf{z})$ such that a descent algorithm on Equation 9 from a constant starting point (in $\mathbf{z}$), yields a minima $\mathbf{z}_{\mathrm{map}}$ and $\mathbf{x}_{\mathrm{map}} = \mathbf{Ez}_{\mathrm{map}}$.*

Theorem 3.1 is a special case of the mountain bypass theorem, that is presented in Appendix A, along with its proof. We now demonstrate the importance of this Theorem using a simple example.

**Example 3.2. Pointwise recovery (denoising) with a double potential prior**
*Assume that $x \in \mathbb{R}$, and the forward mapping is the identity. That is, $b = x + \epsilon$, where $\epsilon$ is a small Gaussian noise. This is the simplest 1-dimensional denoising problem. Assume that the prior of $x$ is a double well potential of the form*

$$\pi(x) \propto \exp\left(-\gamma^{-1}(x - \mu)^2(x + \mu)^2\right).$$

*This potential is plotted in Figure 3(a). Given data b, for the MAP estimator, one needs to minimize*

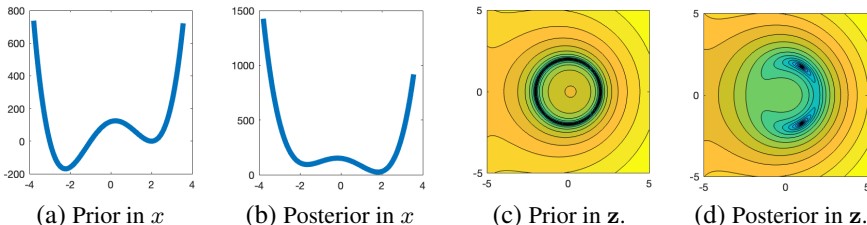

(a) Prior in $x$      (b) Posterior in $x$      (c) Prior in $\mathbf{z}$.      (d) Posterior in $\mathbf{z}$.

Figure 3: A non-convex prior with 2 local minima in $x$ is replaced with a learned quasi-convex prior in higher dimensions. Plots (c) and (d) are in log scale.

*the log posterior distribution and solve the optimization problem*

$$\min_x \frac{1}{2\sigma}(x - b)^2 + \gamma^{-1}(x - \mu)^2(x + \mu)^2.$$

*This is clearly a non-convex problem. The negative log posterior is plotted in Figure 3(b). This problem has two local minima. One close to $\mu$ and the other close to $-\mu$. A gradient descent type algorithm would therefore converge to one of the two minima, depending on the starting point.*

*Now, consider the simple embedding $x = \mathbf{E}\mathbf{z}$ where $\mathbf{E} = [1, 0]$ and $\mathbf{z} = [z_1, z_2]^\top$. Let us explore Equation 9 where we can learn or choose a function $\phi(\mathbf{z})$ as we please. One such function is*

$$\phi(\mathbf{z}) = (z_1^2 + z_2^2 - \mu^2)^2 = \left((z_1 - \mu)(z_1 + \mu) + z_2^2\right)^2$$

*that is plotted in Figure 3(c). Note that for $z_2 = 0$ the prior is reduced to the original prior. The function in 2D has a clear path that connects both minima in 1D. The posterior surface of $\mathbf{z}$ given the data is plotted in Figure 3(d). The function also has 2 minima however, they are benign since both of them have the same $z_1 = x$ and therefore upon solving this problem with a gradient descent method, we can obtain the unique and correct minima.*

We materialize the descent of Equation 9 using a network of the form:

$$\textbf{OPTEnet}: \quad \mathbf{z}_{j+1} \;=\; \mathbf{z}_j - h_j \mathbf{E}^\top \mathbf{A}^\top(\mathbf{A}\mathbf{E}\mathbf{z}_j - \mathbf{b}) - h_j \boldsymbol{\nabla}_{\mathbf{z}}\phi(\mathbf{z}_j; \boldsymbol{\theta}) \tag{10}$$

where $h_j$ is a non-negative step size. We name this network OPTEnet since it evolves from an *optimization* setting on the hidden Embedded variable $\mathbf{z}$. The method bears similarity to the method proposed by Jin et al. (2017) for solving inverse problems. However, it differs from (Jin et al., 2017) in its utilization of a learnable embedding matrix $\mathbf{E}$. Note, that the network has a single learnable embedding matrix $\mathbf{E}$ and a single potential layer $\phi(\cdot; \boldsymbol{\theta})$, that is shared across all layers, parameterized by the weights $\boldsymbol{\theta}$.

However, using a single embedding and shared parameters $\boldsymbol{\theta}$ may yield a network with limited expressiveness (Ongie et al., 2020), as it is shared across all layers. One way to increase the expressiveness of the network is to unroll the iteration (Ongie et al., 2020), effectively changing both $\mathbf{E}$ and $\boldsymbol{\theta}$ with the iteration $j$, obtaining:

$$\textbf{EUnet}: \quad \mathbf{z}_{j+1} \;=\; \mathbf{z}_j - h_j \mathbf{E}_j^\top \mathbf{A}^\top(\mathbf{A}\mathbf{E}_j\mathbf{z}_j - \mathbf{b}) - h_j \boldsymbol{\nabla}_{\mathbf{z}}\phi(\mathbf{z}_j; \boldsymbol{\theta}_j) \tag{11}$$

We call this network EUnet as it includes an *unrolling* and embedding steps. This network extends the idea of embedding beyond a single embedding matrix. While Equation 10 is a gradient flow step for the optimization problem in Equation 9, its unrolled variant in Equation 11 is not.

**Stochastic sampling.** Note that Equation 10, and Equation 11 are deterministic discretizations of differential equations. However, similarly to (Martin et al., 2012; Yang et al., 2022; Croitoru et al., 2023), it is possible to augment Equation 10, and Equation 11 with a stochastic sampling mechanism as shown in Equation 6, yields a stochastic differential equation. This extension of our model is left for future work.

**Embedding and Potential Implementation.** The networks in Equation 10 and Equation 11 require a choice of a potential function for $\mathbf{z}$ and a selection of embedding matrices $\mathbf{E}$. We provide a detailed description of their particular implementation in Appendix B.

## 4 NUMERICAL EXPERIMENTS

As we see next, the methods proposed in this work perform well for classical problems such as tomography and image deblurring, similarly to other existing methods. More importantly, our proposed method significantly outperforms existing methods for highly ill-posed problems such as the inversion of magnetic data. In our experiments, we use we use the MNIST (Lecun & Cortes), and STL10 (Coates et al., 2011) datasets.

We experiment with the two variants of the embedded solution proposed in this paper, namely **OPTnet** and **EUnet**. As a comparison, we consider a diffusion model applied to an inverse problem as proposed by Chung et al. (2022a) that is denoted by 'Diffusion' throughout this section, and the Unrolled proximal iteration proposed by Mardani et al. (2018), similarly denoted by 'Proximal'. We discuss training and evaluation details in Appendix C.

### 4.1 THE DUALTHLON PROBLEM

In Section 2 we have seen how diffusion models converge to two different local minima in Figure 1(b) - one which is local and is unrealistic and one is global and is therefore desired. As can be seen in Figure 1(c), using the same data with our **OPTEnet**, we obtain a sampling of the correct minima.

We now do a similar test on a large number of points. We first train the networks **OPTEnet** and **EUnet** to solve the problem. The embedding dimension of $\mathbf{z}$ here is 128. Thus, it is 64 times larger than the size of the input $\mathbf{x}$. We also train a network based on proximal methods (Mardani et al., 2018) where no embedding is used for comparison.

We generate our training and validation data by sampling 10,000 data points chosen from 3 Gaussians with means $\mu_i, i = 1, ..., 3,$, that is, $\mathbf{x} \sim N(\mu_i, \mathbf{I})$. For each sampled point $\mathbf{x} = (x_1, x_2)^\top$ we generate the measured data by the summation $x_1 + x_2$, and add 1% noise, attempting to recover $\mathbf{x} = (x_1, x_2)^\top$.

A table with the mean-squared-error (MSE) for each method, on the final data set is presented in Table 1. We find that the proposed **OPTEnet** and **EUnet** architectures, that use embedding of the solution in a larger space, perform significantly better than the inversion based on the proximal method – that solves the problem in the original space. Among the two variants, the Unrolled method EUnet, performed significantly better compared with the optimization-based network OPTEnet.

| Method | Proximal | OPTEnet | EUnet |
|---|---|---|---|
| MSE | $3.5 \times 10^{-1}$ | $8.2 \times 10^{-2}$ | $4.1 \times 10^{-2}$ |

Table 1: Mean-Square-Error on validation data for the duathlon problem

### 4.2 IMAGE DEBLURRING

Image deblurring is a common inverse problem where the forward problem is given by the integral (see Nagy & Hansen (2006)

$$\mathbf{b}(\mathbf{r}) = \int_\Omega K(\mathbf{r} - \mathbf{r}')\mathbf{x}(r')d\mathbf{r}' \tag{12}$$

where $K(\mathbf{r} - \mathbf{r}')$ is a point spread function (PSF) with the form $K(\mathbf{r} - \mathbf{r}') = \exp\left(-s^{-1}\|\mathbf{r} - \mathbf{r}'\|\right)$. Here $s$ plays the role of smoothing. For a small $s$ the data is very similar to the original image and the problem is almost well posed, while for a large $s$ the data is highly smoothed and the problem is highly ill-posed. We use the algorithm presented in Nagy & Hansen (2006) and discretize the integral on a $96 \times 96$ grid. We then train our network as well as the network Adler & Öktem (2017) on image deblurring problems where the blurring kernel changes from light to heavy blurring. We also use a trained diffusion model as proposed in Song et al. (2022) on recovering the original image. The results are summarized in Table 2.

It is no surprise that our EUnet performs better than the Proximal method, since our method generalizes it. When comparing with diffusion models our method gives slightly worse results for problems

| Blurring Kernel size $s$ | Diffusion | Proximal | EUnet (Ours) |
|:---:|:---:|:---:|:---:|
| 1 | 4.3e-3 | 5.6e-3 | 1.3e-3 |
| 3 | 4.3e-2 | 3.0e-2 | 2.3e-2 |
| 5 | 1.9e-1 | 4.7e-2 | 4.2e-2 |
| 7 | 5.0e-1 | 6.3e-2 | 5.7e-2 |
| 9 | 9.5e-1 | 8.9e-2 | 7.1e-2 |

Table 2: Recovery loss (MSE) of the STL-10 test set, of different methods and different blurring kernels sizes $s$ .

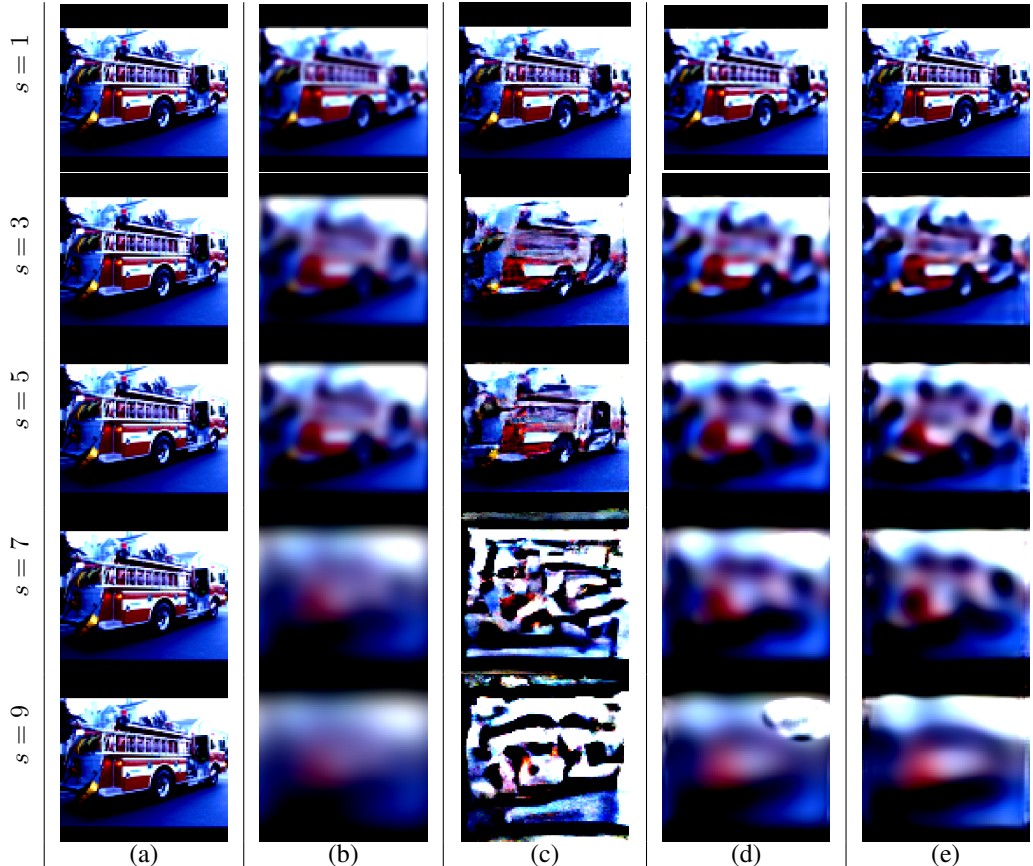

Figure 4: An example of the recovery of deblurred images from the STL-10 data set. (a) Ground truth (same for all rows) (b) Observed data, (c) Diffusion, (d) Proximal, (e) EUnet. Table 2 reports numerical recovery results. Additional examples are provided in Appendix D.

where the blurring is small. Nonetheless, for problems where the blurring is significant, our method easily outperforms diffusion models.

### 4.3 MAGNETICS

Observing that the real advantage of embedding is obtained for highly ill-posed problems we now turn our attention to such a problem. Magnetics is a classical inverse problem in geophysical exploration Parker (1994) and is commonly used to study the earth's interior and other planets Mittelholz & Johnson (2022). The forward problem is given by Fredholm integral equation of the first kind

$$\mathbf{b}(\mathbf{r}') = \int_\Omega \left(\mathbf{n}_I \cdot \boldsymbol{\nabla}\boldsymbol{\nabla}(|\mathbf{r} - \mathbf{r}'|^{-1}) \cdot \mathbf{n}_J\right) \mathbf{x}(\mathbf{r})d\mathbf{r} \tag{13}$$

where $\mathbf{n}_I$ and $\mathbf{n}_J$ are known direction vectors and $\mathbf{r}$ and $\mathbf{r}'$ are location vectors. The model $\mathbf{x}(\mathbf{r})$ is the magnetic susceptibility to be computed from the data $\mathbf{b}$. Typically, the data is measured on the top of the earth and one wishes to estimate the solution everywhere. The magnetic problem is clearly highly ill-posed as we require the recovery of 2D solution from 1D data.

Upon discretization of the integral Equation 13 using the midpoint method, we obtain a linear system. A sample from the images and the data which is a 1D vector that corresponds the forward problem is presented in Figure 5. We use the MNIST data set to train the system using our method as well as diffusion as proposed in Chung et al. (2022a). The true images, images that are generated using the unrolled network EUnet, and images generated by the Diffusion model are presented in Figure 5. The images clearly demonstrate that while the diffusion model fails to converge to an acceptable result, our EUnet yields plausible solutions. For the magnetic problem which is highly ill-posed,

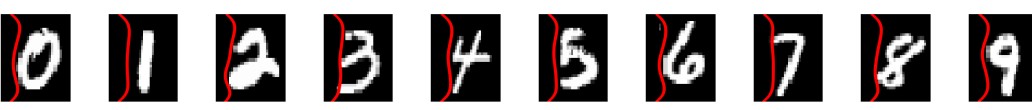

Figure 5: Models and corresponding data (red line) for the inverse magnetics experiment.

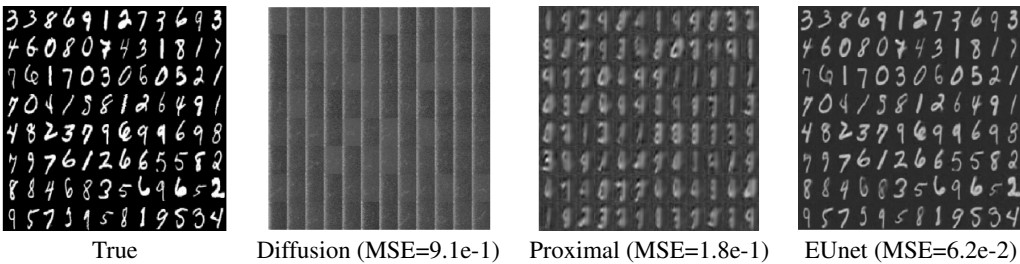

|  True | Diffusion (MSE=9.1e-1) | Proximal (MSE=1.8e-1) | EUnet (MSE=6.2e-2) |

Figure 6: Recovery of MNIST images from magnetic data using Diffusion, Proximal, and EUnet.

we observe that embedding is a key for a successful recovery of the solution. In Appendix D, we show the convergence of each of the methods, showing that our EUnet also offers faster convergence compared with proximal methods, as well as our optimization problem based OPTEnet.

## 5 SUMMARY AND CONCLUSIONS

In this paper, we have introduced a new method for inverse problems that incorporates learning an over-complete embedding of the solution as well as a regularization that acts on the hidden variables. Our method can be used either as an optimization problem or as an unrolled version. Our methods extend techniques that are based on over-complete dictionaries such as basis pursuit by allowing to tailor a data-driven regularization for the basis and extend the regularization learning by changing the basis.

We provide theoretical justification to the method and conduct experiments with a few model problems to demonstrate that indeed, there is merit in jointly learning the embedding as well as the regularization that acts on it.

Comparing our network to existing methods we observe that it outperforms other end-to-end techniques that do not embed the solution in high dimension. Moreover, our method significantly outperforms diffusion-based methods.

We believe that the main reason is that diffusion-based methods are not trained end-to-end and therefore may sample low-energy regions of the posterior. Incorporating an embedding in diffusion models and training them end-to-end is an open question that we believe can improve their performance for highly ill-posed problems.

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

## A  APPENDIX: THE MOUNTAIN BYPASS THEOREM

The difficulty in finding a feasible solution to the inverse problem when working with the original variables $\mathbf{x}$ (specifically in low input dimensions such as RGB images), stems from the fact that the prior, $\pi(\mathbf{x})$ can be highly disjoint. This implies that the posterior contains many local minima and therefore convergence from a fixed initial point, $\mathbf{x}_0$) is not guaranteed. This is because the initial point may be located close to a different local minimum. It is well known (Strang, 1991) that assuming that the function is continuous, there must be a mountain pass between local minima, which hinders the use of gradient descent methods. However, as stated in Theorem 3.1, embedding the solution in high dimensions (effectively replacing $\mathbf{x}$ by $\mathbf{z}$ in our notations) it is possible to converge from a fixed starting point, by designing an appropriate embedding and learning a regularization in this high dimension. We now give theoretical justification to this approach.

Assume that we start the search for $\mathbf{x}$ close to one of the local minima. In order to find any other minima, we cannot use a descent method. When considering non-convex problems a highly cited theorem is the mountain pass theorem.

**The mountain pass problem.** Let $R(\mathbf{x})$ be a bounded function from $\mathbb{R}^N$ to $\mathbb{R}$ with continuous second derivatives. Let $\mathbf{x}_1$ and $\mathbf{x}_2$ be local minima of the function, that is

$$\boldsymbol{\nabla} R(\mathbf{x}_i) = 0 \quad \text{and} \quad \boldsymbol{\nabla}^2 R(\mathbf{x}_i) > 0, \quad i = 1, 2.$$

Then, there exists a path (a mountain pass) defined by

$$\begin{aligned} \mathbf{x}(t) &= \mathbf{x}_1 + \mathbf{s}(t) \\ \mathbf{s}(0) &= 0, \quad \mathbf{s}(1) = \mathbf{x}_2 - \mathbf{x}_1 \end{aligned} \tag{14}$$

such that the function on this path $R(\mathbf{x}(t))$ have a unique maximum.

The mountain pass theorem is a simple extension of Rolle's theorem in basic calculus. It states that in order to move from one minima of a function to another one has to climb a hill.

The mountain pass theorem (Strang, 1991) assumes that we are limited to the topography of the function $R(\mathbf{x})$. We now consider a theorem that can be used in order to learn minimizers that do not need to climb a hill in order to get to a lower minimum.

**Theorem A.1.** *(The Mountain Bypass).* Let $\mathbf{x}_1$ and $\mathbf{x}_2$ be local minima of $R(\mathbf{x})$ with $R(\mathbf{x}_2) \le R(\mathbf{x}_1)$. Assume some embedding of the form

$$\mathbf{x} = \mathbf{E}\mathbf{z}$$

where $\mathbf{E}$ is an $n \times k$ full rank matrix with $k > n$. Assume also that there is a function, $g : R^n \to R^k$ such that $g(\mathbf{x}) = \mathbf{z}$, that uniquely maps the vectors $\mathbf{x} \in R^n$ onto a subspace of the vectors in $R^k$.

Finally, consider a function $\Psi : R^k \to R$ such that

$$\Psi(g(\mathbf{x})) = R(\mathbf{x})$$

Then, there exist an embedding $\mathbf{E}$ and a function $\Psi$ such that if $\mathbf{x}_1$ and $\mathbf{x}_2$ are local minima of $R$ and $\mathbf{z}_1 = g(\mathbf{x}_1)$ and $\mathbf{z}_2 = g(\mathbf{x}_2)$ then there is a continuous path $\mathbf{z}(t), 0 \le t \le 1$ such that $\mathbf{z}(0) = \mathbf{z}_1$, $\mathbf{z}(1) = \mathbf{z}_2$ and

$$\frac{d\Psi(\mathbf{z}(t))}{dt} \le 0, \quad 0 \le t \le 1$$

That is, there is a path $\mathbf{z}(t)$ that bypass the mountain pass when we are unlimited by the topography given in the space spanned by $\mathbf{x}$.

**Proof:** *We prove the theorem by a simple construction. We choose $g(\mathbf{x}) = \mathbf{X}\mathbf{x}$ where $\mathbf{X} = \mathbf{E}^\dagger$. We can then decompose $\mathbf{z}$ into two parts, one in the active space of $\mathbf{E}$ and one in the null space of $\mathbf{E}$, that is*

$$\mathbf{z} = \mathbf{X}\mathbf{x} + \mathbf{Y}\mathbf{y} \tag{15}$$

*where the matrices $\mathbf{X}$ and $\mathbf{Y}$ are the active and null space of $\mathbf{E}$, that is*

$$\mathbf{EX} = \mathbf{I} \quad \mathbf{EY} = 0 \quad \text{and} \quad \mathbf{Y}^\top \mathbf{Y} = \mathbf{I}$$

*Given a vector $\mathbf{z}$ one could compute $\mathbf{x}$ and $\mathbf{y}$ by $\mathbf{x} = \mathbf{Ez}$ and $\mathbf{y} = \mathbf{Y}^\top \mathbf{z}$.*

*We then propose to construct $\Psi$ from two parts*

$$\Psi(\mathbf{z}) = R(\mathbf{x}) + \Omega(\mathbf{y}) = R(\mathbf{Ez}) + \Omega(\mathbf{Y}^\top \mathbf{z}) \tag{16}$$

*where $\Omega(0) = 0$, is a continuously differentiable function that we can choose as we please.*

*Let $\mathbf{z}_1 = \mathbf{X}\mathbf{x}_1$ and $\mathbf{z}_2 = \mathbf{X}\mathbf{x}_2$. Clearly, neither $\mathbf{z}_1$ or $\mathbf{z}_2$ has a component in the null space of $\mathbf{E}$.*

*To build a path from $\mathbf{z}_1$ to $\mathbf{z}_2$ we first consider the part in the active space of $\mathbf{E}$. We set the active part of the space as the mountain pass Equation 14*

$$\mathbf{x}(t) = \mathbf{x}_1 + \mathbf{s}(t)$$

*Next, we consider a path in the orthogonal part of $\mathbf{E}$ and let*

$$\mathbf{y}(t) = t(1 - t)\mathbf{y}$$

*for some $\mathbf{y} \neq 0$.*

*We then have that*

$$\Psi(\mathbf{z}(t)) = R(\mathbf{x}(t)) + \Omega(\mathbf{y}(t)).$$

*Since we are free to choose $\Omega$ as we please we choose it such that $\Psi(\mathbf{z}(t))$ is monotonically decreasing for $0 \leq t \leq 1$.*

In the context of solving the inverse problem, the problem the original problem with $\mathbf{x}$ is changed to the problem

$$\min_{\mathbf{z}} \frac{1}{2}\|\mathbf{AEz} - \mathbf{b}\|^2 + R(\mathbf{Ez}) + \Omega(\mathbf{Y}^\top \mathbf{z})$$

The addition of a learnable function $\Omega$ is the one that allows us to move from one minimum to the next without the need to go through the mountain pass. In the learning process proposed above, rather than learning $R$ and then $\Omega$ we choose a function $\phi(\mathbf{z})$ that yields a smooth path from the starting point $\mathbf{z}_1$ to the desired point, $\mathbf{z}_2$ that is the solution to the inverse problem.

## B    APPENDIX: EMBEDDING AND NETWORK ARCHITECTURES

The systems in Equations 10 and 11 require the utilization of a gradient of the potential function $\phi$. Note that in contrast to any other inversion technique known to us, our $\phi$ operates on the higher dimension $\mathbf{z}$ rather on the lower dimension $\mathbf{x}$.

While it is possible to use many different architectures for the potential our basic form for the potential $\phi$ is a simple network of the form

$$\phi(\mathbf{z}, \boldsymbol{\theta}) = \mathbf{q}^\top \sigma_I \left( \mathbf{K}_n \sigma_I (\mathbf{K}_{n-1} \sigma_I (... \mathbf{K}_2 \sigma_I (\mathbf{K}_1 \mathbf{z} + t\mathbf{b})) \right) \tag{17}$$

Here, the learnable parameters are $\boldsymbol{\theta} = \{\mathbf{K}_1, \ldots, \mathbf{K}_N, \mathbf{b}, \mathbf{q}\}$, where $\mathbf{b}$ is the time embedding. The activation function $\sigma_I$ are chosen as the integral of common activation functions. For example, in our experiments, we choose the integral of the hyperbolic tangent ($\tanh$) activation function:

$$\sigma_I(t) = \int \tanh(t)\, dt = \log(\cosh(t)).$$

Note that upon differentiating we obtain

$$\boldsymbol{\nabla}_{\mathbf{z}}\phi(\mathbf{z}, \boldsymbol{\theta}) = \mathbf{K}_1^\top \operatorname{diag}(\sigma(\mathbf{K}_1 \mathbf{z} + b)) ... \mathbf{K}_{n-1}^\top \operatorname{diag}(\sigma(\mathbf{K}_{n-1}\mathbf{a}_{n-1}))\mathbf{K}_n^\top \operatorname{diag}(\sigma(\mathbf{K}_n \mathbf{a}_n))\mathbf{q} \tag{18}$$

$$\mathbf{a}_j = \sigma_I(\mathbf{K}_{j-1}\sigma_I(... \mathbf{K}_2 \sigma_I(\mathbf{K}_1 \mathbf{z} + t\mathbf{b}) \tag{19}$$

For data (e.g. duathlon problem) we choose $\mathbf{K}_i$ as dense layers and for image data, we choose them as $s \times s$ convolutions. Note that our choice of activation makes the network a composition of convex functions and therefore, for $\mathbf{q} \geq 0$ the network is convex with respect to $\mathbf{z}$.

In order to learn the embedding matrix $\mathbf{E}$ we use a simple network that takes $t$ and maps it into a kernel using a 2 layer network, that is

$$\mathbf{E}(t) = \mathbf{W}_2 \sigma(t\mathbf{W}_1 + \mathbf{c}) \tag{20}$$

Here $\mathbf{W}_1, \mathbf{W}_2$ and $\mathbf{c}$ are trainable weight matrices that determine the embedding as a function of time.

Note that is in Equation 17 we use a single Layer with $\mathbf{K}_1 = \alpha\mathbf{I}$ with $\alpha \gg 1$ and we use $\mathbf{E}$ to be a Haar transform then we obtain the well-known $\ell_1$ regularization of the wavelet transform. If on the other hand, we reduce $\mathbf{E}$ to the identity matrix and choose $\phi$ to be a Unet then we obtain an architecture similar to the one proposed in Adler & Öktem (2017). Thus, our architecture allows for the combination and extension of both techniques.

## C    TRAINING AND EVALUATION DETAILS

Throughout the experiments in Section 4, we aim to train the 'Proximal', OPTEnet, and EUnet to reduce the residual, i.e., to decrease the data fit error. Denoting the noisy data by $\mathbf{b}$, and the true solution as $\mathbf{x}$, we aim to minimize the empirical risk:

$$\mathcal{L} = \mathbb{E}\|f(\mathbf{b}) - \mathbf{x}\|_2, \tag{21}$$

where $f(\cdot)$ is the respective neural network.

We train each network for 300 epochs, and determine the hyperparameters by a grid search. Our hyperparameters are the learning rate $\mathrm{lr} \in \{1e-2, 1e-3, 1e-4, 1e-5\}$, the weight decay $\mathrm{wd} \in \{1e-3, 1e-4, 1e-5, 1e-6, 0\}$ and batch size $\mathrm{bs} \in \{32, 64, 128, 256, 512\}$.

To train the diffusion model (Chung et al., 2022a), we used the code provided by the authors. Note that as per the work suggested in (Chung et al., 2022a) a major difference between the diffusion model and 'Proximal' (Mardani et al., 2018) and our OPTEnet and EUnet, is that the diffusion model is not trained in an end-to-end manner for the specific inverse problem we aim to solve. As can be seen in our experiments, the approach of learning a 'global' prior via a diffusion model, may lead to sub-par results on hard inverse problems, where the forward problem is not close to denosing.

## D    ADDITIONAL VISUALIZATIONS AND CONVERGENCE PLOT

**Additional deblurring visualizations.** now provide additional visualizations of the recovery quality of our EUnet compared with Proximal methods (Mardani et al., 2018) and Diffusion models Chung et al. (2022a). As is evident from Table 2, our EUnet offers better recovery (lower MSE), and this improvement is reflected in the provided examples below.

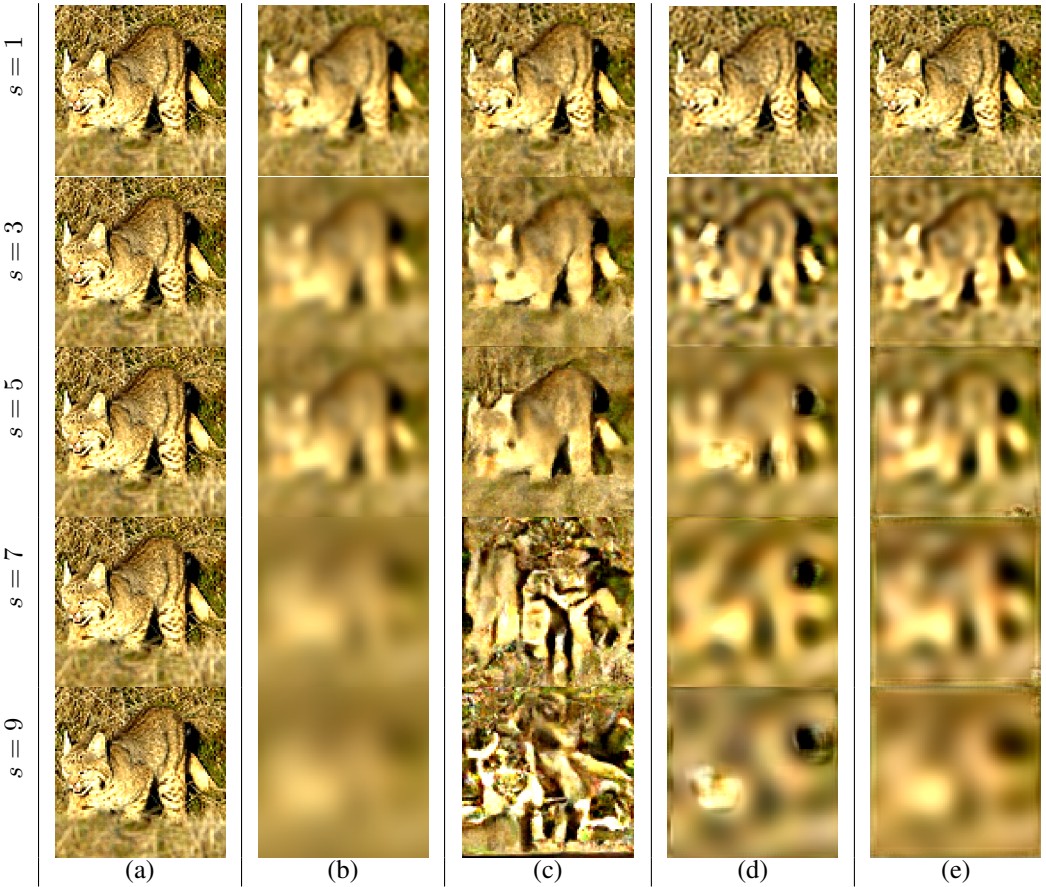

Figure 7: An additional example of the recovery of deblurred images from the STL-10 data set. (a) Ground truth (same for all rows) (b) Observed data, (c) Diffusion, (d) Proximal, (e) EUnet. Table 2 reports numerical recovery results.

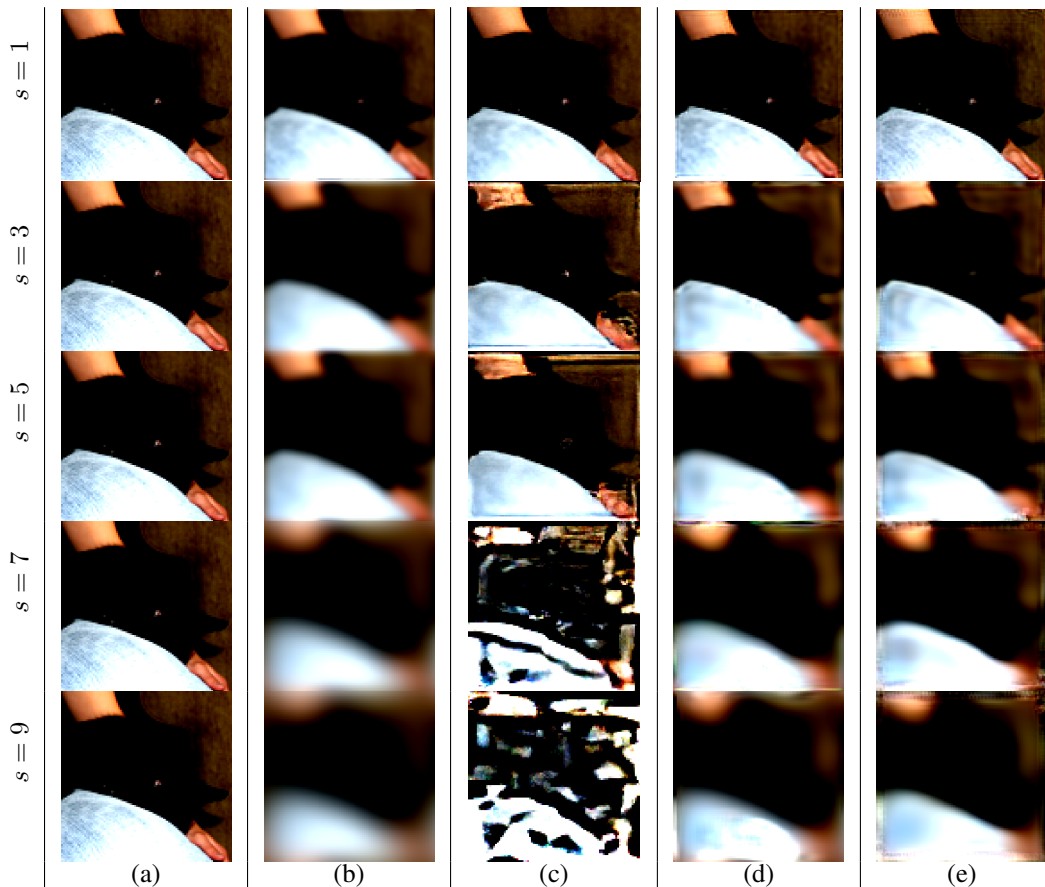

Figure 8: Additional example of the recovery of deblurred images from the STL-10 data set. (a) Ground truth (same for all rows) (b) Observed data, (c) Diffusion, (d) Proximal, (e) EUnet. Table 2 reports numerical recovery results.

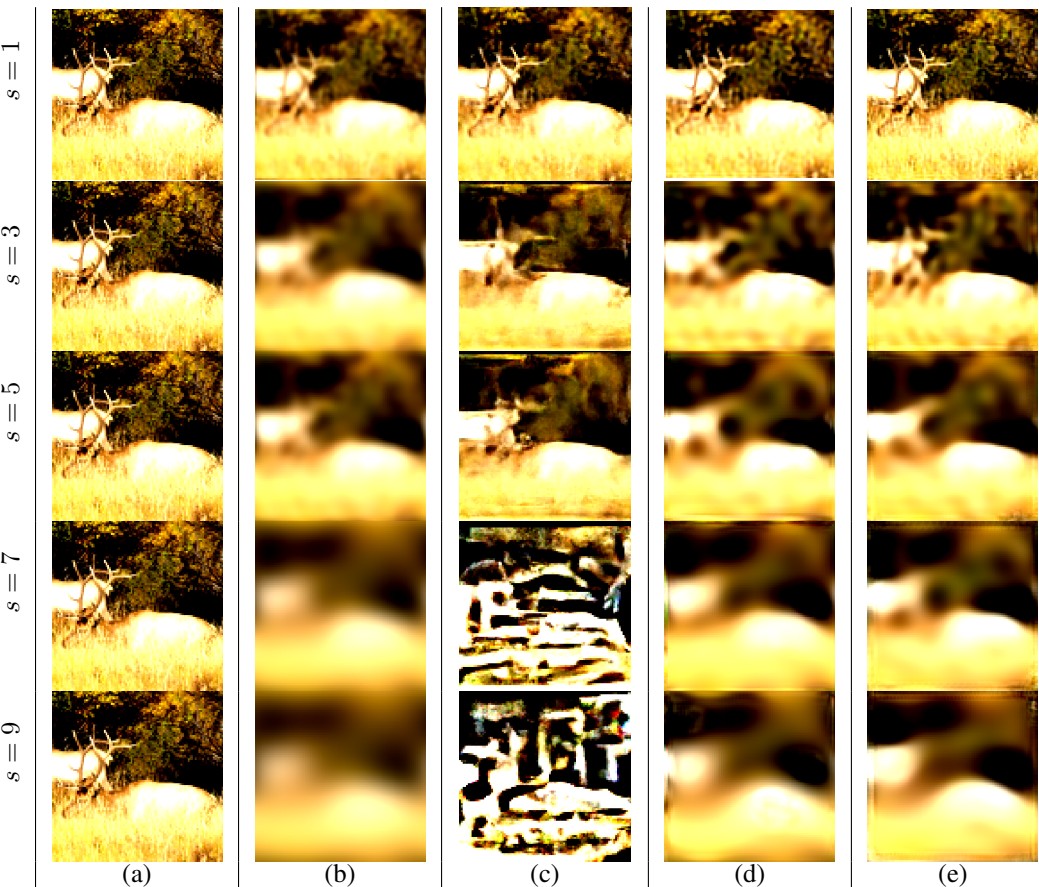

Figure 9: An additional example of the recovery of deblurred images from the STL-10 data set. (a) Ground truth (same for all rows) (b) Observed data, (c) Diffusion, (d) Proximal, (e) EUnet. Table 2 reports numerical recovery results.

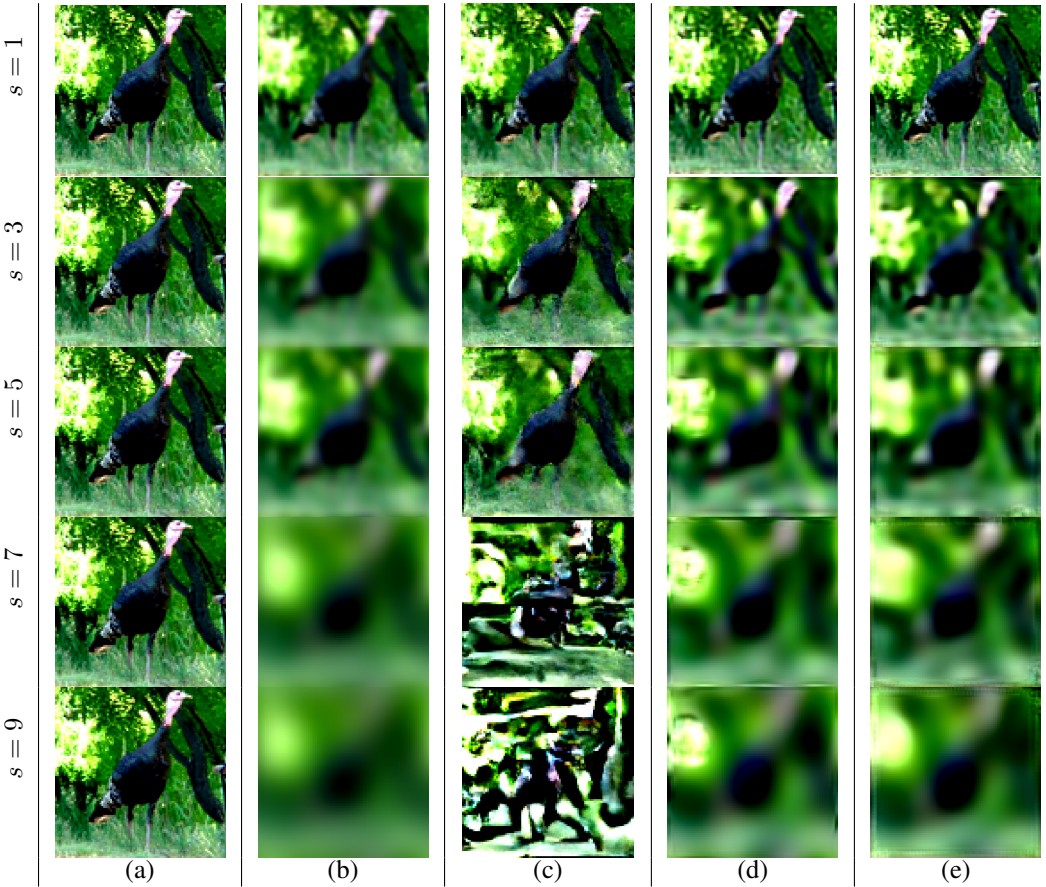

Figure 10: Additional example of the recovery of deblurred images from the STL-10 data set. (a) Ground truth (same for all rows) (b) Observed data, (c) Diffusion, (d) Proximal, (e) EUnet. Table 2 reports numerical recovery results.

**Convergence of EUnet.** We report the convergence plots of the Proximal Mardani et al. (2018) method and our OPTEnet and EUnet, in Figure 11. We observe that our EUnet offers faster convergence, in addition to the improved performance in terms of recovery.

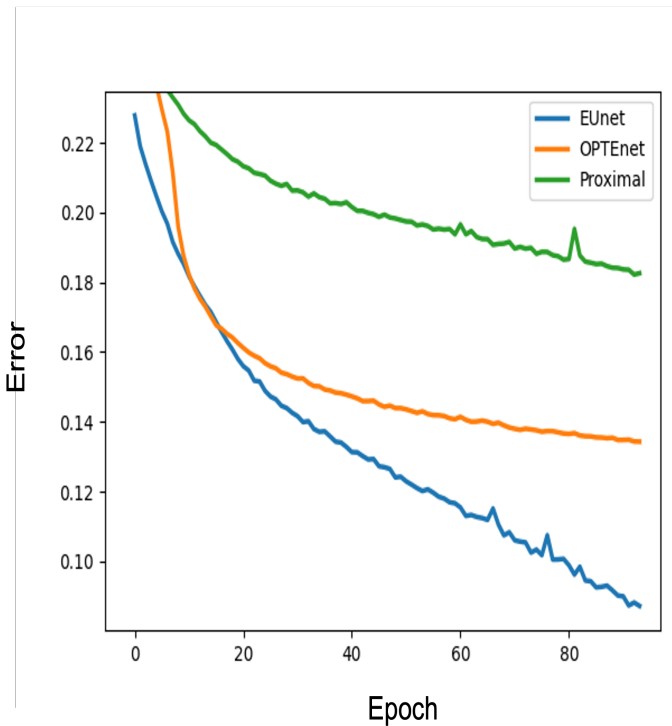

Figure 11: Convergence history for different methods in the magnetics experiment.

