# OpenReview forum: "Embedding Improves Neural Regularizers for Inverse Problems"
_ICLR.cc/2024/Conference — Submitted to ICLR 2024_

### Official Review · Reviewer_riDU · 2023-10-30

**Soundness:** 2 fair
**Presentation:** 3 good
**Contribution:** 2 fair
**Rating:** 3
**Confidence:** 3

**Summary:**

An often-utilized approach for addressing the linear inverse problem $A x + \epsilon = b$ is to solve the regularized least squares problem (or Maximum A Posteriori, MAP):

$$ \min_{x'} \| A x' - b \|_2 + \psi(x') $$

for some (convex) regularizer $\psi$.

Inspired by the mountain pass theorem this paper proposes to

a) consider the extended problem $$ \min_{z'} \| A E z' - b \|_2 + \phi(z') $$ where E is a short and wide matrix.

b) unroll the iteration of the gradient descent method, learning both the embedding E and the regularizer $\phi$

The paper concludes with experiments validating the proposed approach.

**Strengths:**

The paper is well-written and includes informative toy examples that illustrate the intuition behind their approach.

**Weaknesses:**

- The proposed approach appears to be a minor variation of the unrolling methods that have been extensively studied in the past. It is possible that similar techniques may have been previously presented elsewhere.

- The experimental section of the paper is relatively weak. Given the experimental nature of the paper, it would be beneficial to compare the proposed method with more recent supervised techniques, offering more quantitative results. This could include metrics such as the average performance on a subset of the test set, providing a more robust evaluation.

- Additionally, comparing the proposed approach with the DPS method seems unfair, as DPS operates in a distinct unsupervised regime where no target values are provided. Furthermore, it appears from observations, such as Figure 4, that DPS outperforms the proposed approach when the problem is expected to be solvable. For instance, when $s = 9$, it is conceivable that a simpler method, like a direct estimation of $x$ from $b$, might yield better (or comparable) results than the proposed approach.

- Lastly, the experiments do not clearly demonstrate whether the role of the embedding is merely to increase the number of parameters in the neural networks, which is a well-established practice. A more convincing set of experiments would compare the proposed method with the one without embedding *but* keeping the number of learnable parameters fixed.

**Questions:**

- Regarding Theorem 3, it seems that the proof shows only that the map $\psi$ decreases through the path, not that the whole loss function (including the data misfit term) decreases. Please clarify.

- The experiments related to Figure 6 are unclear. Is the same line/measurement used for the same digit class? How this lines are chosen and why are they chosen to avoid the digits?

- Figure 11 is this training error?

---

### Official Review · Reviewer_zbc8 · 2023-10-30

**Soundness:** 3 good
**Presentation:** 2 fair
**Contribution:** 2 fair
**Rating:** 3
**Confidence:** 2

**Summary:**

This article provides another approach to learning a dictionary and a regularizer simultaneously for a linear inverse problem.

**Strengths:**

The article is relatively verbosely and clearly written. It addresses a linear inverse problem, which appears in many places across the sciences.

**Weaknesses:**

- Overall, I am not convinced at the utility of this approach to
solve basic linear problems over existing methods. Linear inverse
problems have been solved well repeatedly for decades. The writing
focuses on some very
basic dynamical systems, and does not demonstrate a significant
improvement over existing deblurring methods. The case for learning
the regularizer and the embedding simultaneously, while
theoreticaly interesting, is not made sufficiently strong from the
empirical evidence. For instance, does this approach even produce better
results than a simple linear Wiener filter? Also, there is not much theoretical
contribution. So, in my opinion, this does not substantiate a sufficient amount
 of contribution.



- These experiments are not particularly convincing to me. The
  authors do not discuss the amount of computational time (or
floating point operations, or other alternative) required by each
method. Many proximal methods (especially for poorly conditioned
blurring kernels) take a very long time to converge; I am not
convinced that the images are the true solutions found by the
proximal method. Also, the diffusion model *should* have some sort
of convergence guarantee, and the spurious recoveries (with large
black and white spots) look like it must have come from a numerical
error.

**Questions:**

Does learning the regularizer and dictionary simultaneously come with an increased cost?

How does this method compare (in both computational cost and quality of solutions) to the (expansive) amount of other approaches for linear inverse problems?

---

### Official Review · Reviewer_9vsv · 2023-10-30

**Soundness:** 2 fair
**Presentation:** 3 good
**Contribution:** 3 good
**Rating:** 5
**Confidence:** 4

**Summary:**

This paper applies over-complete dictionary learning ideas to develop learning-based solutions to imaging inverse problems. In particular, it proposes solving inverse problems using an unrolled iterative algorithm that gradually reconstructs the image by solving for the coefficients of a (learned) overcomplete dictionary while imposing a (learned) prior on that overcomplete dictionary.

The proposed method is tested on an optimization problem in R^2, image deblurring, and reconstructing MNIST digits from simulated "magnetics" measurements. The proposed method outperforms related baseline methods.

**Strengths:**

Generally well-written.

Proposed method is interesting and (somewhat) novel.

**Weaknesses:**

## Validation:
Comparisons aren't comprehensive---the proposed method is only compared to two simple algorithms and the experiments are peformed at low resolution.
Validated primarily on toy (MNIST) datas.

## Exposition:
The duathalon toy example isn't entirely convincing and deserves further explanation. Increasing the penalty on the data fidelity term used in the reconstruction process would avoid landing on the second modes the authors point to as a limitation of existing methods.

## Minor organization issue:
I suggest moving Fig 1 up a line above "Exampl 2.1. ...". The way it breaks up the text is confusing at the moment (looks like the end of the page).

## Statements without evidence:
In reference to replacing simple TV priors with learned probability density estimates, the paper says "As we see next, this makes the estimation of the model x substantially more difficult, because we need to derive algorithms that avoid local minima, and to find the global minima of the neural regularize R(x)."


## Minor misrepresentation of the literature:
The manuscript cites 3 2022 papers for learning independent priors for inverse problems. Learning independent priors for inverse problems has been widely used since at least 2013; see [A] and the hundreds of papers (many learning-based) that have built upon it.
[A] Venkatakrishnan, Singanallur V., Charles A. Bouman, and Brendt Wohlberg. "Plug-and-play priors for model based reconstruction." 2013 IEEE global conference on signal and information processing. IEEE, 2013.

## Typos:
Pg 4: "obfuscate these difficulties". I think you meant obviate
Fig 3: Please label (c) and (d) axis.

**Questions:**

"Typically, the data is measured on the top of the earth and one wishes to estimate the solution everywhere. The magnetic problem is clearly highly ill-posed as we require the recovery of 2D solution from 1D data." Is this a simplification of a 2D to 3D problem?

Learned linear dictionaries can work well on MNIST. How does the proposed method compare to a fixed sparse dictionary?

How important is unrolling for the method?

Does the method scale to higher resolutions?

## Other suggestions:
Perhaps report table 2 results in terms of PSNR.

---

### Official Review · Reviewer_TUQu · 2023-11-06

**Soundness:** 2 fair
**Presentation:** 2 fair
**Contribution:** 1 poor
**Rating:** 3
**Confidence:** 4

**Summary:**

This paper aims at solving inverse problems by embedding the solution into higher dimensions (i.e. transforming/representing the solution space of the inverse problems into a higher-dimensional space) and jointly learning the embedding matrix/operator and its regularizer. Indeed, the paper explores the problem of sparse coding via algorithm unrolling where both the embedding matrix (sparsifying frame or dictionary) and regularizer are jointly learned.

**Strengths:**

1) The paper explores a crucial approach for addressing inverse problems.

2) The outcomes achieved using the proposed method surpass those obtained by the diffusion-based model.

**Weaknesses:**

1) Lack of (or at very least ambiguity) in the contribution

2) Lack of proper review of the related works

3) Lack of discussion regarding theoretical convergence guarantees, and conditions for accurate recovery are not discussed

4) The visual presentation of the results lacks engaging or visually compelling elements.

5) The paper doesn't detail the computational overhead or potential limitations of implementing the proposed method in practical scenarios.

**Questions:**

The paper in question would benefit from addressing several issues:

    1. Throughout the paper, the terms “embedding”, “embedding the solution into higher dimensions”, “embedded solution vector”, and “embedding the solution in a larger space” have been repeated while no definition for each of them is provided. This work lacks of clear definitions for the terminology “embedding” which could hinder comprehension, especially for readers not familiar with these concepts. Since, this work shares several similarities with ‘sparse coding’, thus it is advisable to elucidate on such terms in a way that a reader from other field (especially those who are familiar with the concept of sparsity) would grasp the terms.

    2. Needs improved referencing, particularly in clarifying the use of terms and concepts similar to ‘sparse coding’ and in referencing relevant works such as Tikhonov's work for $\ell_2$ smoothness regularization.

    3. citing (Tenorio et al., 2011) in this paper is not needed, and it can be removed.

    4. The third paragraph on page 1 should be revised to cover three prevalent methodologies commonly employed for tackling linear inverse problems, i.e. end-to-end deep learning-based approaches, unrolled algorithms, and the plug-and-play prior framework (which employs denoisers as implicit priors). It's important to note that while these are significant approaches, they don't encompass the entirety of solving methods for such problems. Proper referencing and detailing of these methodologies will provide a comprehensive overview for the readers.

    5. A subsection should also be included to discuss similar/related work, e.g., “Theoretical linear convergence of unfolded ISTA and its practical weights and thresholds”; “ALISTA: analytic weights are as good as learned weights in LISTA”; “Sparse coding with gated learned ISTA”; “Ada-lista: learned solvers adaptive to varying models”; “Neurally augmented ALISTA”.

    6. The third paragraph on page 1 could be rewritten so that it covers three widely-used approaches for solving linear inverse problems with proper references, i.e. end-to-end deep learning-based methods, unrolled algorithms, and plug-and-play prior framework (or using denoiser as implicit priors) -- note that this is not the ultimate category of the solving approaches.

    7. On page 2, in “the unrolling allows for greater expressiveness” and “the original coordinates of the solution”, ‘expressiveness’ and ‘coordinate’ are vague. On page 4, “similarly to other existing methods.” is vague! They require better elaboration or clarification.

    8. Equations (12) and (13) ought to be reformulated into a matrix-vector form that aligns with the structure of the forward model elucidated in Equation (1).

    9. It lacks discussion regarding theoretical convergence guarantees and a comparative analysis for the results without embedding in Figure 6.

    10.  Equations (8) and (9) are pertinent for analysis-based recovery, which, as per compressive sensing theory, can be achieved under the assumption of sparsity or compressibility of $z$ and incoherence between $A$ and $E$. Are these conditions met in the given scenario?

    11.  The visual representation of the results, exemplified in Figures 4, 7-10, demonstrates that the performance of the proposed method is comparable to that of the rival method, specifically, the proximal methods. How do you explain/interpret it?

    12. what does blurring kernel size $s=1$ mean in Table 2?

---

### Author Response · Authors · 2023-11-21
**Thank you to the reviewers**

We thank the reviewers for taking the time to read our paper and for their insightful suggestions.

We will make sure to implement them in our next submission.

With kind regards,

The authors.

---

### Meta-Review · Area_Chair_SpkP · 2023-12-08

**Metareview:**

The reviewers unanimously recommend rejection (3-5-3-3). Major concerns have been raised about the presentation, the relevance, and the evaluation of the approach. The authors acknowledge the concerns and will implement the necessary changes in a major revision submitted to a future conference.

**Justification For Why Not Higher Score:**

The reviewers unanimously recommend rejection.

**Justification For Why Not Lower Score:**

N/A

---

### Decision · Program_Chairs · 2024-01-16

Reject